# Older adults and stroke survivors are steadier when gazing down

Yogev Koren[1,2]*, Shirley Handelzalts[1,2], Yisrael Parmet[3], Simona Bar-Haim[1,2,4]

**1** Physical Therapy Department, Ben-Gurion University of the Negev, Be'er-Sheva, Israel, **2** Translational Neurorehabilitation Laboratory, Ofakim, Israel, **3** Industrial Engineering and Management Department, Ben-Gurion University of the Negev, Be'er-Sheva, Israel, **4** Zlotowski Center for Neuroscience, Ben Gurion University of the Negev, Be'er-Sheva, Israel

\* yogevk@post.bgu.ac.il

## Abstract

### Background

Advanced age and brain damage have been reported to increase the propensity to gaze down while walking, a behavior that is thought to enhance stability through anticipatory stepping control. Recently, downward gazing (DWG) has been shown to enhance postural steadiness in healthy adults, suggesting that it can also support stability through a feedback control mechanism. These results have been speculated to be the consequence of the altered visual flow when gazing down. The main objective of this cross-sectional, exploratory study was to investigate whether DWG also enhances postural control in older adults and stroke survivors, and whether such effect is altered with aging and brain damage.

### Methods

Posturography of older adults and stroke survivors, performing a total of 500 trials, was tested under varying gaze conditions and compared with a cohort of healthy young adults (375 trials). To test the involvement of the visual system we performed spectral analysis and compared the changes in the relative power between gaze conditions.

### Results

Reduction in postural sway was observed when participants gazed down 1 and 3 meters ahead whereas DWG towards the toes decreased steadiness. These effects were unmodulated by age but were modulated by stroke. The relative power in the spectral band associated with visual feedback was significantly reduced when visual input was unavailable (eyes-closed condition) but was unaffected by the different DWG conditions.

### Conclusions

Like young adults, older adults and stroke survivors better control their postural sway when gazing down a few steps ahead, but extreme DWG can impair this ability, especially in people with stroke.

**Data Availability Statement:** All relevant data are within the paper and its Supporting Information files.

**Funding:** The author YK and SH disclosed receipt of the following financial support for the research, authorship, and/or publication of this article: This research was supported by the Helmsley Charitable Trust through the Agricultural, Biological and Cognitive Robotics Initiative and by the Marcus Endowment Fund, both at Ben-Gurion University of the Negev (YK), and by the Israeli Ministry of Science & Technology (SH). The funders had no role in study design, data collection and analysis, decision to publish, or preparation of the manuscript. There was no additional external funding received for this study.

**Competing interests:** The authors have declared that no competing interests exist.

## Background

In clinical settings, it has often been observed that unstable walkers, such as older adults [1] and stroke survivors [2, 3], tend to gaze down while walking. This propensity can be manifested by an inability to walk without gazing down even when instructed not to do so [4]. This tendency is also manifested by a shorter look-ahead distance [5] and/or a lower head angle [6, 7] while walking and by an increase in time spent looking down (e.g., [8]).

In healthy adults, downward gazing (DWG) has been observed when hazards on the walking surface compromise walking stability, such as when negotiating obstacles [9, 10], ascending and descending a stair or stairs [5, 11], walking on uneven surfaces [12] and complex terrain [13, 14], but also when step precision was set as a goal [15, 16].

From these reports in healthy adults, it would be reasonable to speculate that DWG is used in the visuomotor control of stepping. Specifically, the walker attempts to identify safe footholds and/or acquire exproprioceptive information [6] to guide and plan subsequent steps [17–20]. The greater propensity of unstable walkers to gaze down may thus indicate that these individuals require more time for such anticipatory stepping control [5, 8]. Yet, unstable walkers have been reported to gaze down even when walking on a flat, obstacle-free surface [7, 21], where visuomotor control of stepping is redundant. Such observations might indicate that these walkers shift to a more conscious state of control, a shift that has been reported to stem directly from fear of falling [22, 23], but also indirectly from motoric and/or sensory deficits [24].

Recently, Koren et al. [25] proposed an additional interpretation to DWG behavior in unstable walkers. These authors have shown that gazing down enhances standing and walking postural steadiness. They suggested that gazing down changes the visual structure of the environment. This change can affect the visual flow created by the relative motion of the observer and the environment, and since this flow is used for feedback postural control (e.g., [26–28]), the adequacy of the signal for this purpose may change as well. If so, then DWG might serve more than just one purpose/function: both anticipatory stepping control and feedback postural control. However, the results reported by Koren et al. [25] were from a group of only healthy adults. Furthermore, reports about the effect of DWG on postural control are few and inconsistent [29–31].

Therefore, in the current study we aimed to investigate the effect of DWG on standing postural steadiness in older adults and stroke survivors, and to assess whether this effect is altered by aging and/or stroke. To investigate the effect of DWG on postural control, we tested older adults and stroke survivors under the same standing conditions reported by Koren et al. [25]. To assess the effect of aging and stroke, we used data from healthy adults, collected in the above-mentioned report (freely available here), and compared it with the data collected herein. Our hypotheses were that gazing down a few steps ahead would increase standing postural steadiness, as was previously observed in healthy adults, and that older adults and stroke survivors would demonstrate reduced steadiness compared to healthy younger adults. No other assumptions were made.

## Methods

### Participants

Posturography of young adults (20–41 years of age), older adults (65–80 years of age), and stroke survivors (no age restrictions) was evaluated and compared in this experiment. Exclusion criteria included orthopedic or neurological conditions that significantly affect gait (e.g., joint replacement, Parkinson's Disease). Participants with common age-related conditions

such as diabetes were permitted to participate. For older adult and stroke participants a score of $\geq 24$ in the Mini-Mental State Exam was required in order to participate in the study. All participants were able to ambulate without assistance. Participants provided written consent and were given monetary compensation to cover travel expenses. This study complies with the standards of the Declaration of Helsinki and was approved by the Ethics Committee at Shamir Medical Center, Be'er Yaakov, Israel (MOH_2018-02-14_002188).

Participants were tested under five visual conditions: eyes closed (EC), downward gazing at their feet (DWGF), downward gazing one meter ahead (DWG1), downward gazing three meters ahead (DWG3), and forward gazing (FG) at a target located approximately 4.2 meters ahead at eye level.

## Testing conditions and procedure

Participants were instructed to stand barefoot, as still as possible, on a Kistler 9286AA force platform (Kistler Instrument Corp., Winterthur, Switzerland) in a standardized stance, i.e., with their feet tight together and hands loosely hanging at their sides. The ability to maintain this stance was determined with their eyes closed before testing. Participants unable to maintain this stance for 30 seconds were tested in a wide-base stance (i.e., heels 6 cm apart and feet rotated outwards at a 10° angle from midline).

Five 30-second quiet-standing trials in each of the five visual conditions were performed (with a total of 25 trials in random order). Raw data from the force plate were collected, at 100Hz, using a data acquisition system consisting of a data acquisition box (Kistler A/D type 5691) and Bioware (version 5.3.0.7) software.

Before each trial (i.e., a single 30-sec stand) the force plate was calibrated with no weight (i.e., participants were instructed to step off the platform). Following the calibration, participants were instructed to stand on the platform and continuously look at one of the four targets. Locations for DWG1, DWG3, and FG were marked with coloured circles 20 cm in diameter. For the DWGF, participants were instructed to look at their own toes, while for the EC condition, participants were just instructed to close their eyes.

## Data processing and outcome measures

The recordings were then processed using a dedicated MATLAB (MathWorks Inc. version R2016b) script. First, the center of pressure (COP) time series was low-passed using a 2nd-order Butterworth filter with a cut-off frequency of 15Hz, and the first three seconds and the last second were removed from the time series. As the main outcome measure, the script computes the short-term diffusion coefficient ($D_s$) of COP derived from stabilogram diffusion analysis (SDA) as described by Collins and De-Luca [32]. Briefly, the diffusion coefficient is the rate at which the quadratic Euclidean distance between two COP positions increases as a function of the time interval between them. In this experiment we calculated three coefficients: the coefficient for sway as a 2D motion ($D_{rs}$), and two coefficients for its 1D components— anterior-posterior ($D_{ys}$) and medio-lateral ($D_{xs}$)—all given in $mm^2$/sec. The short-term diffusion coefficients were reported to be reliable [32], and more sensitive than COP-based summary statistics [33]; also, because they quantify the dynamic nature of steady-state stance, these coefficients can be more informative regarding the underlying control mechanism. In general, smaller values signify increased steadiness and are usually considered as indication of better postural control, while larger values are thought to be indicative of decreased steadiness due to impaired control.

As a secondary outcome measure, the script also computes the relative power in the frequency band 0.01–0.1 Hz, as described by Singh et al. [34]. Specifically, after zero padding to

10,000 data points (to achieve a resolution of 0.01 Hz) and de-meaning, we used Fast Fourier Transform to obtain the single-sided power spectra of the COP time series. The power spectrum was normalized to the total power in the signal, and the relative power in the 0.01–0.1 Hz band was calculated. When visual information was unavailable (compared to when it was) [34], the relative power of this frequency band significantly decreased, indicating to a visual contribution to postural control in this frequency band. To investigate whether the visual system plays any role in the effect of DWG on postural sway, we wanted to test whether values would differ between conditions.

## Sample size estimation

To estimate the number of participants required in each group, to show an effect of DWG on postural sway, we used the data collected from the younger adults group. We used the 'SIMR' package [35] in R (Version 4.0.5), in conjunction with the 'lme4' package [36]. This package allows users to calculate power for generalized linear mixed models. The power calculations are based on Monte Carlo simulations [35]. We simulated multiple experiments with only DWG3 and FG as levels of the fixed effect, at various levels of the random effect (i.e., number of participants). The predicted term in these simulations was the variable Drs. The results of these simulations revealed that the observed power reached ~90% with only five participants. Taking into account that this effect may be smaller in older adults and stroke survivors, we doubled this number and set the sample at 10 participants in each group.

## Statistical analysis

For statistical analyses we included all data from the force platform (i.e., every trial was a point of measurement). In two cases we had missing data due to technical issues, and in one case the participant performed only four repetitions of each condition. Missing values were replaced with the average of the participant in the condition. Since the sway parameters' distribution significantly deviated from normal, we used a logarithmic transformation (denoted as Ln(*parameter*)). The transformed values were analysed using linear mixed-effect models, with *Condition*, *Group*, and their interaction as *fixed* effects, and participant as the *random* effect. Statistical significance was set a priori at $\alpha < 0.05$; sequential Bonferroni corrections were applied for multiple comparisons when appropriate. Analysis was performed using SPSS (v.26, IBM Corp, Armonk, USA).

# Results

Ten older adults and ten stroke survivors participated in this study, performing a total of 500 trials. Data from 15 young participants who had performed an additional 375 trials, the results of which were previously published [25], were also included in the analysis. Descriptive statistics of the demographic data of these participants are reported in Table 1.

## Main results

The main results of all models are presented in the (see S1 Table).

The model for Drs (planar sway) revealed significant main effects for the condition ($F_{4,860}$ = 163.8, p<0.001) and the group ($F_{2,860}$ = 6.4, p = 0.002), and a significant interaction term ($F_{8,860}$ = 1.98, p = 0.046). Post-hoc pairwise comparisons between the different conditions are presented in Fig 1A. Briefly, pairwise comparisons revealed that sway values did not differ between the DWGF and FG conditions (p = 0.14), and neither between the DWG1 and DWG3 conditions (p = 0.48), in which minimal values were observed. Mean values of all other pairs were significantly different from one another ($p < 10^{-6}$). Comparing between groups (see

**Table 1. Descriptive statistics and demographic data.**

| | younger | | older | | stroke | |
|---|---|---|---|---|---|---|
| | mean | range | mean | range | mean | range |
| Age $_{years}$ | 28*^ | 20–41 | 72^ | 66–79 | 63 | 50–73 |
| Height $_{cm}$ | 170 | 154–193 | 164 | 145–179 | 165 | 154–178 |
| Mass $_{kg}$ | 69.5^ | 51–94 | 77 | 52–103 | 84 | 56–113 |
| BMI $^{k}g/_{m^2}$ | 24*^ | 20–31 | 28 | 24–34 | 31 | 22–40 |
| Visual acuity | | | 0.57 | 0.32–0.63 | 0.55 | 0.32–0.8 |
| | count | | count | | count | |
| Base $_{narrow/wide}$ | 15/0 | | 9/1 | | 7/3 | |
| Glasses $_{no/yes}$ | 6/9 | | 0/10 | | 2/8 | |
| Sex $_{male/female}$ | 7/8 | | 5/5 | | 6/4 | |

The * indicates a significant difference from the older adults' group, and the ^ indicates a significant difference from the stroke survivors' group. Visual acuity (EU decimal system) of younger adults is missing since they were not tested for it.

Fig 1B) revealed that mean sway value of the younger adults was lower than those of the older adults (contrast estimates = -0.47, p = 0.02) and the stroke survivors (contrast estimate = -0.62, p = 0.003), but mean sway values did not differ between the older adults and the stroke survivors (contrast estimate = -0.14, p = 0.48).

Exploring the interaction term revealed that the younger and older adults were affected similarly by the different visual conditions (consistent with the main effect of the condition), but the stroke survivors were affected differently. (See details below.)

The model for Dys (1D sway in the AP) revealed significant main effects for the condition ($F_{4,860}$ = 135.5, p<0.001) and for the group ($F_{2,860}$ = 6.6, p = 0.001), but the interaction term was non-significant ($F_{8,860}$ = 1.2, p = 0.32). Post-hoc pairwise comparisons between the different conditions are presented in Fig 2A. Briefly, mean sway value in the DWG3 condition was

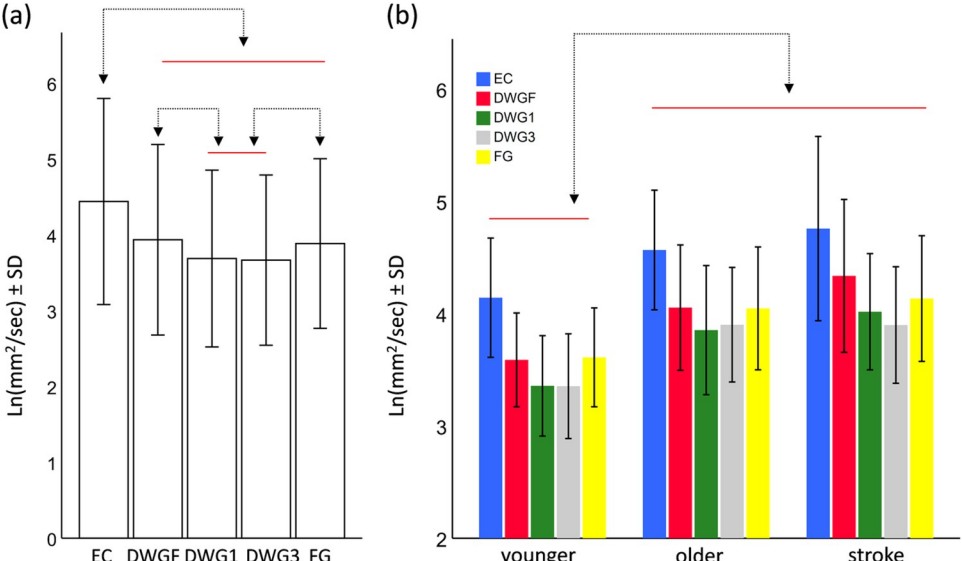

**Fig 1. Pairwise comparisons of the main effect of the Condition (a) and the main effect of the Group (b), for the parameter Drs.** Horizontal bars are grouping elements, and arrows indicate a significant difference at the level of α<0.05. Abbreviations: eyes closed (EC), downward gazing to feet (DWGF), downward gazing one meter ahead (DWG1), downward gazing three meters ahead (DWG3), and forward gazing (FG).

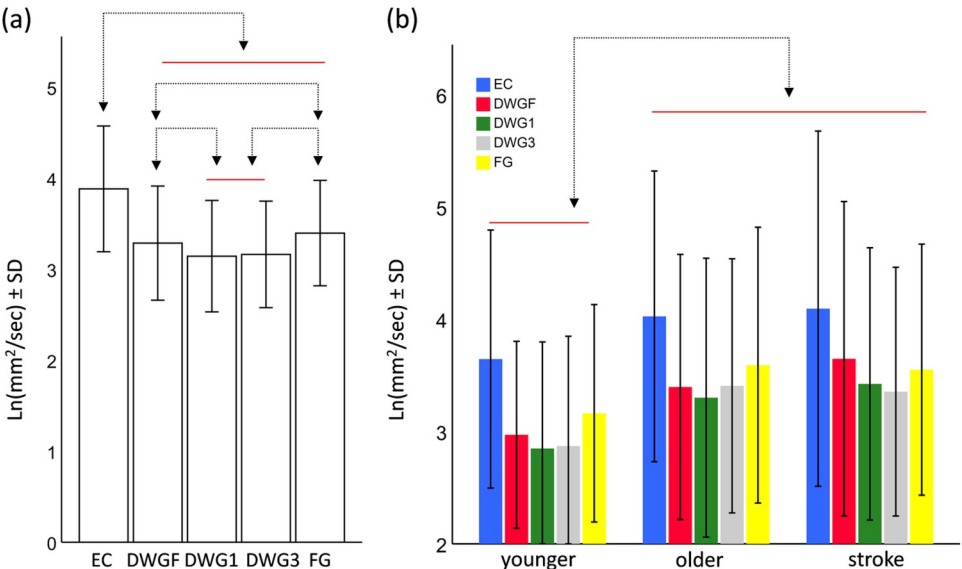

**Fig 2. Pairwise comparisons of the main effect of the Condition (a) and the main effect of the Group (b), for the parameter Dys.** Horizontal bars are grouping elements, and arrows indicate a significant difference at the level of α<0.05. Abbreviations: eyes closed (EC), downward gazing to feet (DWGF), downward gazing one meter ahead (DWG1), downward gazing three meters ahead (DWG3), and forward gazing (FG).

significantly lower than in all other conditions. All pairwise comparisons revealed significant differences. The main effect of the group indicated that mean sway value of the younger adults was lower than those of the older adults (contrast estimates = -0.47, p = 0.04) and the stroke survivors (contrast estimate = -0.71, p = 0.001), but mean sway values did not differ between the older adults and the stroke survivors (contrast estimate = -0.24, p = 0.28).

The model for Dxs (1D sway in the ML) revealed significant main effects for the condition ($F_{4,860}$ = 127.2, p<0.001) and for the group ($F_{2,860}$ = 4.3, p = 0.014), and a significant interaction term ($F_{8,860}$ = 2.4, p = 0.014). Post-hoc pairwise comparisons between the different conditions are presented in Fig 3A. Briefly, minimal and equivalent sway values were observed in the DWG1 and DWG3 condition. All other comparisons revealed significant differences. The main effect of the group (see Fig 3B) indicated that mean sway value of the younger adults was lower than those of the older adults (contrast estimates = -0.44, p = 0.048) and the stroke survivors (contrast estimate = -0.52, p = 0.027), but mean sway values did not differ between the older adults and the stroke survivors (contrast estimate = -0.07, p = 0.74).

Exploring the interaction term revealed that the younger and older adults were affected similarly by the different visual conditions, but the stroke survivors were affected differently.

From Figs 1B and 3B it seems that the significant interaction term, observed in the Drs and Dxs models, might be a consequence of the observed effect of the stroke group in the DWGF condition. That is, only in stroke survivors did the mean sway value in the DWGF condition exceed that of the mean value during the FG condition. To test this possibility, we excluded the DWGF condition from both models, and found the interaction term to be non-significant ($F_{6,688}$ = 1.3, p = 0.24 and $F_{6,688}$ = 1.4, p = 0.23 for Drs and Dxs respectively), confirming our suspicion.

## Secondary results

Finally, for the parameter *relative power* we transformed values by extracting their square-root. The results of this model revealed significant main effects for the condition ($F_{4,860}$ = 6.9,

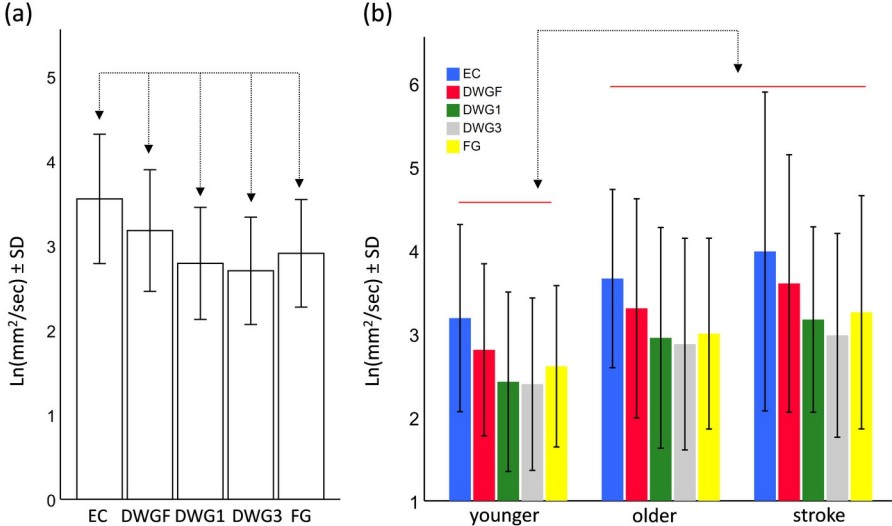

**Fig 3. Pairwise comparisons of the main effect of the Condition (a) and the main effect of the Group (b), for the parameter Dxs.** Horizontal bars are grouping elements, and arrows indicate a significant difference at the level of $\alpha < 0.05$. Abbreviations: eyes closed (EC), downward gazing to feet (DWGF), downward gazing one meter ahead (DWG1), downward gazing three meter ahead (DWG3), and forward gazing (FG).

$p<0.001$) and for the group ($F_{2,860} = 8.1$, $p<0.001$), but not for the interaction term ($F_{8,860} = 1.1$, $p = 0.33$). Pairwise comparisons (see Fig 4) revealed that the relative power in the EC was significantly lower than in all other conditions ($p<0.001$), but all other comparisons revealed non-significant differences. Comparing between groups revealed that the relative power in the younger adults group was greater than in the older adults (contrast estimate = 0.05, $p<0.001$) and the stroke survivors (contrast estimate = 0.04, $p = 0.003$). No difference was observed between the older adults and the stroke survivors (contrast estimate = -0.01, $p = 0.5$)

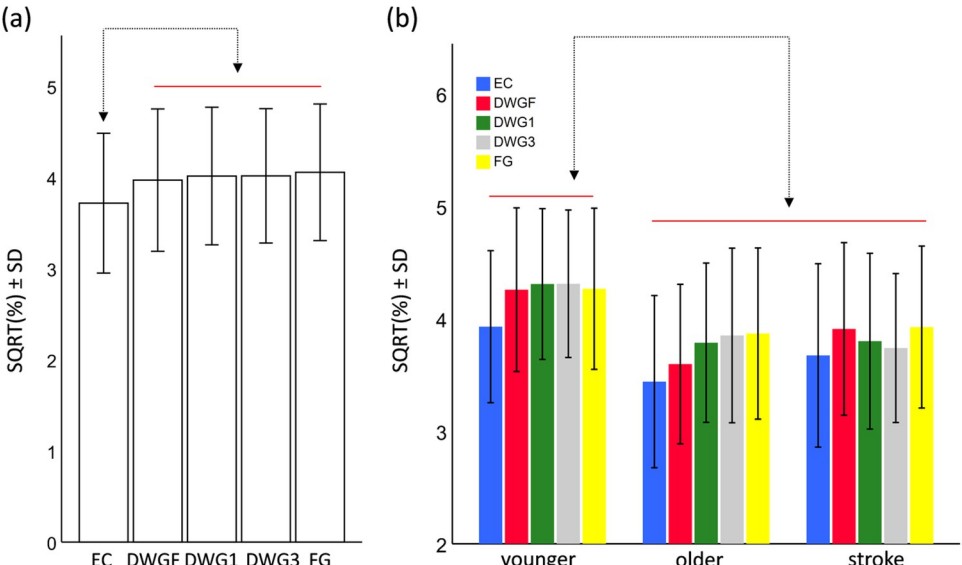

**Fig 4. Pairwise comparisons of the main effect of the Condition (a) and the main effect of the Group (b), for the parameter relative power.** Horizontal bars are grouping elements, and arrows indicate a significant difference at the level of $\alpha < 0.05$. Abbreviations: eyes closed (EC), downward gazing to feet (DWGF), downward gazing one meter ahead (DWG1), downward gazing three meters ahead (DWG3), and forward gazing (FG).

## Discussion

In this study we aimed to investigate whether DWG affects standing postural sway in populations characterized by this gaze behavior, and whether such effect is modulated by aging and stroke. The main finding of the study revealed that gazing down one and three meters ahead decreased body sway in young, older, and stroke participants. Also, a greater body sway was observed in older adults and persons with stroke compared with young adults. Interestingly, we found that the visual conditions affected young and older adults in a similar manner but differently in persons with stroke. Our findings suggest that gazing down a few steps ahead enhances the ability to attenuate postural sway, but extreme DWG (i.e., downward gazing towards the feet) can impair this ability, especially in persons with stroke.

In addition to the above findings, mean sway value in the EC condition was significantly greater than in all eye-open conditions. While this finding is not novel or unique, it is frequently used to signify that visual input is important for postural control. The fact that our findings are consistent with the literature supports the validity of our measurements.

The tested distances for downward gazing (i.e., one and three meters) used in the current study are distances that are commonly used during visually guided walking, [13, 15]. We found that the planar sway (Drs) in the DWG1 and DWG3 conditions was significantly lower in comparison to all other conditions, indicating that downward gazing a few meters ahead enhanced the ability to attenuate body sway. Decomposing sway into its 1D components revealed that this was true for ML sway (Dxs) and for AP sway (Dys), but for the latter significantly lower sway values were observed in the DWG3 condition vs. the DWG1 condition.

While this was not a mechanistic investigation, several mechanisms have been speculated to be involved in the effect of DWG. First, Koren et al. [25] proposed that DWG alters the visual structure of the perceived environment, which can lead to enhanced motion-parallax and visual expansion signals (the two variables of the optic flow that dominate visual control of posture [27, 28, 37]). In an attempt to verify this possibility, we performed spectral analysis and compared the relative power, in a spectral-band thought to be associated with visual feedback [34], among the conditions. While the results of this analysis provided further support that the chosen band is indeed associated with visual information (i.e., the mean value in the EC conditions was significantly lower than in all eyes-open conditions), mean values did not differ among all eyes-open conditions. This may indicate that the mechanism underlying the observed effect of DWG is not related to visual flow or that this parameter is not sensitive enough to identify small changes between these conditions.

Another possibility was proposed by Buckley et al. [29]; these authors controlled the visual structure and found that downward head position (i.e., neck flexion) affected the ability to attenuate body sway. They proposed vestibular and/or biomechanical contribution/s. These authors, however, reported that DWG resulted in increased body sway, not decreased as was observed in this study. Further, Oaki et al. [4] reported that with no visual information (eyes closed), postural sway did not differ between forward gazing and downward gazing, in both older adults and stroke survivors. In fact, these authors found that DWG with eyes open disrupts postural control in older adults but enhances it in stroke survivors. They suggested that a combination of gaze distance and vestibular disfunction might explain these results. Others [30, 31], who reported an effect consistent with our own observations, used only downward eye movement. This may indicate that proprioceptive information from the musculature of the eyes (i.e., extraocular information) might play a role in this effect. Overall, the very few reports and the inconsistency of their outcomes make it hard to determine the mechanism/s underlying the effect of DWG observed herein.

Our second aim was to explore whether the effect of downward gazing on postural steadiness is modulated by age and stroke. Consistent with previous reports [38], our results

indicated greater sway in older adults and stroke survivors than in young adults, suggesting that the ability to attenuate body sway becomes impaired with aging and after stroke. Although our intentions were to separate the effect of aging from brain damage by recruiting participants who were close in age, stroke survivors were significantly younger than older adults in our cohort. In addition, three stroke survivors but only one older adult were tested in a wide-base stance, which is considered less challenging and less sensitive to instability [39]. Thus, it only stands to reason that sway values of participants tested in a wide-base stance reflected an overestimation of their ability and that both the younger age and the greater number of participants tested in a wide-base stance may explain why results for these two groups did not differ.

While we observed similar effects for the different visual conditions in the older and younger adults, indicating that the effect/s observed is not modulated by age, the results for the stroke survivors differed, as can be inferred by the significant interaction term in the Drs and Dxs models. This observation indicates a stroke-related modulation of the effect/s observed. Specifically, only in the stroke group did we observe that the mean value during the DWGF condition was significantly greater than that observed in the FG condition. This difference was confirmed to be the source for the significant interaction term by excluding the DWGF condition from the analysis and observing that the interaction term was no longer significant. This finding indicates that extreme DWG–i.e., downward gazing toward feet—disrupted the ability to attenuate ML body sway in comparison to forward gazing in persons with stroke. While the reason for such an effect might be related to vestibular disfunction following stroke (as suggested by Aoki et al. [4]), it can also be something simple, such as anthropometric differences that are expected in persons with high BMI. That is, with greater abdominal fat one has to shift his/her center of mass backwards to a greater extent, when gazing downward to their feet.

### Limitations

The study has several limitations. First, the motivation to conduct this investigation is primarily derived from the notion that people suffering from walking instability tend to gaze down while walking. This is based on extensive clinical experience but is supported by very limited evidence from the literature. In the current investigation, we did not evaluate the propensity of our participants to gaze down which is likely useful to further support our perspective. Second, in this investigation we specifically targeted two populations that are known to suffer from walking instability, but not necessarily due to the same underlying mechanism. While it was our objective to separate the effect of brain damage from that of aging by recruiting participants of a similar age, our stroke group was significantly younger than the older adult group, limiting our ability to separate the effects of these two factors, a matter that should be further examined in future studies. Third, participants' characteristics did not include any population specific measures, which can be helpful in generalizing our results. This is particularly true for the stroke survivors, for whom characterizing stroke severity, and impairments in the motor and the sensory systems could be useful.

### Supporting information

**S1 File Sway data. This file contains the raw data (untransformed) used for statistical analysis.**
(CSV)

**S1 Table Main results. This file contains a table with the main results of all models.**
(DOCX)

## Author Contributions

**Conceptualization:** Yogev Koren, Simona Bar-Haim.

**Data curation:** Yogev Koren.

**Formal analysis:** Yogev Koren, Shirley Handelzalts, Yisrael Parmet.

**Investigation:** Yogev Koren, Simona Bar-Haim.

**Methodology:** Yogev Koren, Shirley Handelzalts, Yisrael Parmet.

**Resources:** Simona Bar-Haim.

**Supervision:** Simona Bar-Haim.

**Visualization:** Yogev Koren.

**Writing – original draft:** Yogev Koren, Shirley Handelzalts.

**Writing – review & editing:** Yisrael Parmet, Simona Bar-Haim.

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
