## [Decision Letter · Decision Letter 0]

20 Mar 2023

PONE-D-22-23687Older Adults and Stroke Survivors Are Steadier When Gazing DownPLOS ONE

Dear Dr. Koren,

Thank you for submitting your manuscript to PLOS ONE. After careful consideration, we feel that it has merit but does not fully meet PLOS ONE’s publication criteria as it currently stands. Therefore, we invite you to submit a revised version of the manuscript that addresses the points raised during the review process.

We look forward to receiving your revised manuscript.

Kind regards,

Nili Steinberg

Academic Editor

PLOS ONE

Journal Requirements:

“The author YK disclosed receipt of the following financial support for the research, authorship, and/or publication of this article: this research was supported by the Helmsley Charitable Trust through the Agricultural, Biological and Cognitive Robotics Initiative and by the Marcus Endowment Fund, both at Ben-Gurion University of the Negev. The funders had no role in study design, data collection and analysis, decision to publish, or preparation of the manuscript.”

3. We noted in your submission details that a portion of your manuscript may have been presented or published elsewhere. [Data of healthy young adults, from a previous publication by our group, is used in the current manuscript as a control group. Most of the data, acquired with older adults and persons with stroke, is new.] Please clarify whether this [conference proceeding or publication] was peer-reviewed and formally published. If this work was previously peer-reviewed and published, in the cover letter please provide the reason that this work does not constitute dual publication and should be included in the current manuscript.

Reviewers' comments:

Reviewer's Responses to Questions

**Comments to the Author**

1. Is the manuscript technically sound, and do the data support the conclusions?

Reviewer #1: Yes

Reviewer #2: Partly

2. Has the statistical analysis been performed appropriately and rigorously? 

Reviewer #1: Yes

Reviewer #2: Yes

3. Have the authors made all data underlying the findings in their manuscript fully available?

Reviewer #1: Yes

Reviewer #2: No

4. Is the manuscript presented in an intelligible fashion and written in standard English?

Reviewer #1: Yes

Reviewer #2: Yes

5. Review Comments to the Author

Reviewer #1: The paper is very interesting and it discuss a very important topic concerning Older Adults and Stroke Survivors stability when gazing down reducing postural sway. It is very important in conidering basophobia and risk factor to predict future Falls in fragile patients. The topic is particularly interesting expecially for its impact on healthcare system, quality of life of patients and their proxies. Due to the pandemic it is of particular relevance for fragile patiens (https://doi.org/10.3390/app12157934).

The paper is suitable for publication.

I coud just suggest some implementation for the background session:

- consider the impact of physical activity on physical and psychical outcome (10.1016/j.archger.2020.104109 and 10.1016/j.jamda.2019.01.128)

- consider the impact of polymedication on older and mood (10.1017/S1041610217001715 and 10.1007/s40520-018-0893-1)

- consider the role of technology in fall prevention (10.1186/s13063-022-06812-w).

Thanks

Reviewer #2: About:

The study expands on a previous study (Koren 2021) on how gazing at different distances affects postural stability. The previous study focused on healthy subjects, whereas this study includes older adults and stroke survivors and tests if there are differences between the groups and different downward gazing conditions. The authors concluded that older adults and stroke survivors had improved postural stability with downward gazing like young adults. However, gazing too close to the feet causes decreased postural stability, especially in the stroke group. The manuscript is very interesting and well-written. The authors did a great job of reporting any missing data, the statistical tools and packages, determining the appropriate sample size, and discussing the limitations of the study.

1. Technical Sound

The authors did a great job explaining the experimental methods and reporting some demographic information about the groups. However, some beneficial demographic information is missing, such as:

i. Were subjects screened based on their vision? It would be interesting to know their ability to see even with corrective lenses. Did everyone have a 20/20 vision? As people get older, their vision tends to get worse, which could affect their postural stability when looking further ahead. Here is a review supporting this: Saftari, L.N., Kwon, OS. Ageing vision and falls: a review. J Physiol Anthropol 37, 11 (2018). https://doi.org/10.1186/s40101-018-0170-1

ii. It would be interesting to know more details about the people who had a stroke, such as clinical scores. This will give the reader an idea of the severity of their disability if they have any.

Having this information would better support the authors’ claims and increase the replication of the results. If unable to provide the additional information, would this be considered a limitation to the study?

2. Statistical Analysis

a. Lines 200, 218, 238, and 263: I can’t seem to find the symbol, *, on the figures that indicate a significant difference. Do you mean the arrows?

b. The authors could consider putting the F and p scores in a table in the main results.

c. Lines 194: “Exploring the interaction term … (see Figure 1b).” Although Figure 1b does seem to show differences between the stroke and age groups, it doesn’t seem to clearly point out that the interaction term between the age and stroke groups are different. For example, the arrows and horizontal bars point out they are significantly different. Consider, adding a marker to indicate the groups that were affected differently.

3. Data fully available

I can’t seem to find the raw data used for the statistical analysis, other than the reported statistical results. Would the authors consider making the raw data available publicly if it currently is not?

4. Grammar and Spelling

Here are some minor grammar edits to consider:

Abstract:

a. Lines 47-48: Consider replacing “persons with stroke” with “people with stroke”

Background:

b. Lines 62: “From these reports …” what are the specific reports that you are referring to?

c. Lines 87-88: “No assumptions were made regarding the effect of age and stroke” This seems confusing to me. What led the authors to test for age and stroke if they didn’t have any assumptions?

Methods:

d. Lines 123: ‘while for the EC condition, no specific instructions were given besides “close your eyes”’ This can cause confusion since the participants seemed like they were instructed to close their eyes.

e. Lines 133: It seems the three coefficients are missing references. It looks like the reference (Koren 2021) provides the equation to calculate the metrics.

5. Additional comments to consider although may not be necessary

a. Additional figures would be great to include. For example, a figure showing the experiment setup.

b. Lines 270: The abbreviation, DWG, is already defined and can be removed. Similar to the other abbreviations in the discussions.

c. When reading the previous study that inspired this study, it seems like there were some differences that the authors took. For example, walking postural stability didn’t seem to be considered. Were some of the stroke survivors unable to walk? It might be good to explain more about the difference in methods between this study and the previous study. Having more info about the stroke group could help with answering this question as well.

6. PLOS authors have the option to publish the peer review history of their article (what does this mean?). If published, this will include your full peer review and any attached files.

Reviewer #1: No

Reviewer #2: No

---

## [Author Response · Author response to Decision Letter 0]

2 Apr 2023

PONE-D-22-23687

Older Adults and Stroke Survivors Are Steadier When Gazing Down

PLOS ONE

Dear Editor and Reviewers,

We wish to thank you all for the time and effort invested in our manuscript. Below you can find our point-by-point response to your comments. 

Response: We revised the manuscript to meet PLOS ONE’s style requirements, as detailed in the provided links. If any requirements are not met, please specify.

“The author YK disclosed receipt of the following financial support for the research, authorship, and/or publication of this article: this research was supported by the Helmsley Charitable Trust through the Agricultural, Biological and Cognitive Robotics Initiative and by the Marcus Endowment Fund, both at Ben-Gurion University of the Negev. The funders had no role in study design, data collection and analysis, decision to publish, or preparation of the manuscript.”

Response: The amended statement was added to the cover letter in ‘Track changes’.

3. We noted in your submission details that a portion of your manuscript may have been presented or published elsewhere. [Data of healthy young adults, from a previous publication by our group, is used in the current manuscript as a control group. Most of the data, acquired with older adults and persons with stroke, is new.] Please clarify whether this [conference proceeding or publication] was peer-reviewed and formally published. If this work was previously peer-reviewed and published, in the cover letter please provide the reason that this work does not constitute dual publication and should be included in the current manuscript.

Response: The cover letter was amended (in ‘Track Changes’) to include our reason to include previously published data within the current report.

Response: We revised the manuscript to comply.

Reviewers' comments:

5. Review Comments to the Author

Reviewer #1: The paper is very interesting and it discuss a very important topic concerning Older Adults and Stroke Survivors stability when gazing down reducing postural sway. It is very important in conidering basophobia and risk factor to predict future Falls in fragile patients. The topic is particularly interesting expecially for its impact on healthcare system, quality of life of patients and their proxies. Due to the pandemic it is of particular relevance for fragile patiens (https://doi.org/10.3390/app12157934).

The paper is suitable for publication.

I coud just suggest some implementation for the background session:

- consider the impact of physical activity on physical and psychical outcome (10.1016/j.archger.2020.104109 and 10.1016/j.jamda.2019.01.128)

- consider the impact of polymedication on older and mood (10.1017/S1041610217001715 and 10.1007/s40520-018-0893-1)

- consider the role of technology in fall prevention (10.1186/s13063-022-06812-w).

Thanks

Response: Thank you for this comment. The reviewer makes a good point about the importance of fear of falling. Specifically, there is evidence in the literature linking fear of falling with increased conscious control of stepping and DWG. We made some changes in the Introduction section (lines 68-71) to introduce the possibility that fear of falling may lead to DWG, due to such an effect. The other suggestions made by this reviewer are all important and relevant, but we feel these overreach the scope and data collected in this study. 

Reviewer #2: About:

The study expands on a previous study (Koren 2021) on how gazing at different distances affects postural stability. The previous study focused on healthy subjects, whereas this study includes older adults and stroke survivors and tests if there are differences between the groups and different downward gazing conditions. The authors concluded that older adults and stroke survivors had improved postural stability with downward gazing like young adults. However, gazing too close to the feet causes decreased postural stability, especially in the stroke group. The manuscript is very interesting and well-written. The authors did a great job of reporting any missing data, the statistical tools and packages, determining the appropriate sample size, and discussing the limitations of the study.

1. Technical Sound

The authors did a great job explaining the experimental methods and reporting some demographic information about the groups. However, some beneficial demographic information is missing, such as:

i. Were subjects screened based on their vision? It would be interesting to know their ability to see even with corrective lenses. Did everyone have a 20/20 vision? As people get older, their vision tends to get worse, which could affect their postural stability when looking further ahead. Here is a review supporting this: Saftari, L.N., Kwon, OS. Ageing vision and falls: a review. J Physiol Anthropol 37, 11 (2018). https://doi.org/10.1186/s40101-018-0170-1

Response: Participants were indeed tested for their visual acuity (excluding the younger adults which were not tested). We added the summary statistics to Table 1. 

ii. It would be interesting to know more details about the people who had a stroke, such as clinical scores. This will give the reader an idea of the severity of their disability if they have any. Having this information would better support the authors’ claims and increase the replication of the results. If unable to provide the additional information, would this be considered a limitation to the study?

Response: We agree with the reviewer. Unfortunately, we did not include clinical tests for stroke participants, besides the MMSE, in our protocol. We have added this information to the Limitation section (lines 373-376). Inclusion criteria in this study required the ability to walk without assistance. The statement “All participants were able to ambulate without assistance” was added to the manuscript (line 101).

2. Statistical Analysis

a. Lines 200, 218, 238, and 263: I can’t seem to find the symbol, *, on the figures that indicate a significant difference. Do you mean the arrows?

Response: Thank you for this comment. This is our mistake. We corrected this in all the above-mentioned instances (Figs 1-4).

b. The authors could consider putting the F and p scores in a table in the main results.

Response: We have added a Table with the main results as a supplementary file (S2_Table). We added a reference to this table in line 197.

c. Lines 194: “Exploring the interaction term … (see Figure 1b).” Although Figure 1b does seem to show differences between the stroke and age groups, it doesn’t seem to clearly point out that the interaction term between the age and stroke groups are different. For example, the arrows and horizontal bars point out they are significantly different. Consider, adding a marker to indicate the groups that were affected differently.

Response: The reviewer makes a good point as figure 1b includes only the pairwise comparisons between groups (although the mean value in each condition within each group is shown in the figure). We revised the text so that the reference to this figure comes before the description of these comparisons in the text (line 205). The same mistake appears for the parameter Dxs (figure 3b), and this was corrected in the same manner (line 244). We also revised the legends of the associated figures. For both the legend is now: “Pairwise comparisons of the main effect of the Condition (a) and the main effect of the Group (b), for the parameter….”

3. Data fully available

I can’t seem to find the raw data used for the statistical analysis, other than the reported statistical results. Would the authors consider making the raw data available publicly if it currently is not?

Response: Yes, the data will be made available as a supplementary materials file, if the manuscript is accepted for publication.

4. Grammar and Spelling

Here are some minor grammar edits to consider:

Abstract:

a. Lines 47-48: Consider replacing “persons with stroke” with “people with stroke”

Response: We revised the text to comply (line 48).

Background:

b. Lines 62: “From these reports …” what are the specific reports that you are referring to?

Response: The sentence was revised to state “From these reports in healthy adults it would be reasonable…” (line 61)

c. Lines 87-88: “No assumptions were made regarding the effect of age and stroke” This seems confusing to me. What led the authors to test for age and stroke if they didn’t have any assumptions?

Response: We revised the sentence to clarify: “Our hypotheses were that gazing down a few steps ahead would increase standing postural steadiness, as was previously observed in healthy adults, and that older adults and stroke survivors would demonstrate reduced steadiness compared to healthy younger adults. No other assumptions were made.(lines 88-91)

Methods:

d. Lines 123: ‘while for the EC condition, no specific instructions were given besides “close your eyes”’ This can cause confusion since the participants seemed like they were instructed to close their eyes.

Response: The text was revised: “…while for the EC condition, participants were just instructed to close their eyes.” (line 127)

e. Lines 133: It seems the three coefficients are missing references. It looks like the reference (Koren 2021) provides the equation to calculate the metrics.

Response: The 3 coefficients were originally described by Collins and De-Luka [1]. Their paper describes both the way they are calculated and provide a theoretical framework for their meaning. Further, this paper also shows that these are reliable measures, and they also showed [2] that these are more sensitive than summary statistics to detect the effect of aging. This information is provided in the manuscript. (lines 134-142)

[1] Collins JJ, De Luca CJ. Open-loop and closed-loop control of posture: a random walk analysis of center-of-pressure trajectories. Exp Brain Res 1993;95(2):308-318.

[2] Collins JJ, De Luca CJ, Burrows A, Lipsitz LA. Age-related changes in open loop and closed-loop postural control mechanisms. Exp Brain Res 1995;104(3):480-492.

5. Additional comments to consider although may not be necessary

a. Additional figures would be great to include. For example, a figure showing the experiment setup.

Response:We agree with the reviewer that a figure of the experimental setup will be helpful, but we did not take any pictures during the experiment (due to privacy restrictions).

b. Lines 270: The abbreviation, DWG, is already defined and can be removed. Similar to the other abbreviations in the discussions.

Response: Redundant definitions were removed from the Discussion section.

c. When reading the previous study that inspired this study, it seems like there were some differences that the authors took. For example, walking postural stability didn’t seem to be considered. Were some of the stroke survivors unable to walk? It might be good to explain more about the difference in methods between this study and the previous study. Having more info about the stroke group could help with answering this question as well.

Response: Our previous report included two experiments: one for standing and one for walking. Naturally we are especially interested in walking, so we also tested some of the participants while walking. Nevertheless, we still have some technical issues with the walking experiment, so we are unable to present any results at this time. As stated above, all participants in this experiment were able to walk without assistance (added in line 101).

---

## [Decision Letter · Decision Letter 1]

24 Apr 2023

Older adults and stroke survivors are steadier when gazing down

PONE-D-22-23687R1

Dear Dr. Koren,

We’re pleased to inform you that your manuscript has been judged scientifically suitable for publication and will be formally accepted for publication once it meets all outstanding technical requirements.

Kind regards,

Nili Steinberg

Academic Editor

PLOS ONE

Additional Editor Comments (optional):

Reviewers' comments:

Reviewer's Responses to Questions

**Comments to the Author**

1. If the authors have adequately addressed your comments raised in a previous round of review and you feel that this manuscript is now acceptable for publication, you may indicate that here to bypass the “Comments to the Author” section, enter your conflict of interest statement in the “Confidential to Editor” section, and submit your "Accept" recommendation.

Reviewer #1: All comments have been addressed

Reviewer #2: All comments have been addressed

2. Is the manuscript technically sound, and do the data support the conclusions?

Reviewer #1: Yes

Reviewer #2: Yes

3. Has the statistical analysis been performed appropriately and rigorously? 

Reviewer #1: Yes

Reviewer #2: Yes

4. Have the authors made all data underlying the findings in their manuscript fully available?

Reviewer #1: Yes

Reviewer #2: Yes

5. Is the manuscript presented in an intelligible fashion and written in standard English?

Reviewer #1: Yes

Reviewer #2: Yes

6. Review Comments to the Author

Reviewer #1: Authors addressed comments and suggestions or argumented the choice.

The topic is very interesting and important for field.

The article is suitable of publication

Reviewer #2: (No Response)

7. PLOS authors have the option to publish the peer review history of their article (what does this mean?). If published, this will include your full peer review and any attached files.

Reviewer #1: No

Reviewer #2: No

---

## [Editor Report · Acceptance letter]

11 May 2023

PONE-D-22-23687R1 

Older adults and stroke survivors are steadier when gazing down 

Dear Dr. Koren:

I'm pleased to inform you that your manuscript has been deemed suitable for publication in PLOS ONE. Congratulations! Your manuscript is now with our production department. 

Kind regards, 

on behalf of

Prof. Nili Steinberg 

Academic Editor

PLOS ONE